# Antiaging Mechanism of Natural Compounds: Effects on Autophagy and Oxidative Stress

**DOI:** 10.3390/molecules27144396

**Published:** 2022-07-08

**Authors:** Elizabeth Taylor, Yujin Kim, Kaleb Zhang, Lenne Chau, Bao Chieu Nguyen, Srujana Rayalam, Xinyu Wang

**Affiliations:** 1DeBusk College of Osteopathic Medicine, Lincoln Memorial University, Harrogate, TN 37752, USA; elizabeth.taylor02@lmunet.edu; 2Department of Pharmaceutical Sciences, College of Pharmacy, Philadelphia College of Osteopathic Medicine-Georgia Campus, Suwanee, GA 30024, USA; yujinki@pcom.edu (Y.K.); kalebzh37@gmail.com (K.Z.); lc8241@pcom.edu (L.C.); bn207851@pcom.edu (B.C.N.); srujanara@pcom.edu (S.R.)

**Keywords:** aging, oxidative stress, autophagy, CDDO derivatives, CAPE, xanthohumol, guggulsterone, resveratrol, sulforaphane

## Abstract

Aging is a natural biological process that manifests as the progressive loss of function in cells, tissues, and organs. Because mechanisms that are meant to promote cellular longevity tend to decrease in effectiveness with age, it is no surprise that aging presents as a major risk factor for many diseases such as cardiovascular disease, neurodegenerative disorders, cancer, and diabetes. Oxidative stress, an imbalance between the intracellular antioxidant and overproduction of reactive oxygen species, is known to promote the aging process. Autophagy, a major pathway for protein turnover, is considered as one of the hallmarks of aging. Given the progressive physiologic degeneration and increased risk for disease that accompanies aging, many studies have attempted to discover new compounds that may aid in the reversal of the aging process. Here, we summarize the antiaging mechanism of natural or naturally derived synthetic compounds involving oxidative stress and autophagy. These compounds include: 2-cyano-3,12-dioxoolean-1,9-dien-28-oic acid (CDDO) derivatives (synthetic triterpenoids derived from naturally occurring oleanolic acid), caffeic acid phenethyl ester (CAPE, the active ingredient in honey bee propolis), xanthohumol (a prenylated flavonoid identified in the hops plant), guggulsterone (a plant steroid found in the resin of the guggul plant), resveratrol (a natural phenol abundantly found in grape), and sulforaphane (a sulfur-containing compound found in cruciferous vegetables).

## 1. Introduction

Aging is generally defined as the process of growing old, which consists of a deterioration of organelles brought about by an increase in reactive oxygen species and a decrease in autophagy, which is responsible for flushing out damaged cells. However, there is no consensus definition for the term aging, reflecting the lack of understanding of the phenomenon itself. To understand the fundamental mechanisms that regulate aging, the science community came up with the theory of aging: the oxidative stress theory of aging (OSTA) [1]. The OSTA is the most well-known and studied mechanistic theory of aging and suggests that the rate of aging is regulated by the accumulation of oxidative damage. The development of aging is a multifactorial process [2,3]. According to the theory, there are two categories that contribute to the development of chronic diseases and premature aging: the accumulation of molecular and cellular oxidative damages in all tissues throughout life and the disruption of immune response homeostasis, characterized by increased pro-inflammatory stimuli and a decreased anti-inflammatory response [2,4,5]. In addition, research over the last decade has demonstrated that many molecular changes associated with aging are caused by the altered autophagy across diverse species. Autophagy (from the Greek words auto, meaning ‘self’, and phagein, meaning ‘to eat’) is a complex pathway that removes cellular components, such as defective organelles and misfolded proteins, through lysosomes. The effect of autophagy regulation on tissue homeostasis has been actively investigated in recent research, which revealed a complex and multifactorial relationship between autophagy and aging.

In recent years, many researchers aimed that autophagy is a druggable target, and modulating the autophagy processes can be a promising therapeutic opportunity for antiaging effects [6,7,8,9]. Pharmacological agents such as metformin, rapamycin, spermidine, and tomatidine have been extensively studied and reviewed as autophagy inducers that extend the health span and increase the lifespan of laboratory animals [10,11,12,13,14]. However, in terms of promoting longevity and antiaging effects, natural products and phytochemicals without side effects are preferred over pharmacological agents.

Scientific research on the potential for a variety of natural compounds and phytochemicals of health benefits has expanded rapidly over the past decade. Phytochemicals are a group of a wide variety of compounds that occur naturally in plants, which includes fruits, vegetables, grains, and other plant food [15]. Phytochemicals produce activity in biological systems, including humans. They consist of nutrients essential for optimal health (e.g., proteins, carbohydrates, vitamins, and minerals) and other chemicals (e.g., phenolic acids, flavonoids, and other phenolics) [16]. These chemicals in plants play crucial roles in their growth and development. They protect plants from harmful agents, such as insects and microbes, as well as environmental factors, such as ultraviolet irradiation and extreme temperatures [17]. Moreover, a number of these phytochemicals are found to have beneficial health effects when consumed. Preclinical, clinical, and epidemiological research suggests that phytochemicals may be effective in treating various diseases owing to their antioxidant and anti-inflammatory activities [18]. Numerous research groups have shown that phytochemicals work by targeting specific receptors, interrupting disease pathways, and disrupting pathogenic life cycles. Some of these natural compounds have been developed into pharmacological agents such as salicylates found in willow bark, quinine in cinchona bark, and proanthocyanidins in cranberries [17].

About 10,000 different phytochemicals have been identified, and a large percentage of these compounds still remain unknown with respect to chemical structure and/or the biological role in humans [19]. Generally, phytochemicals have been classified into six major categories based on their chemical structures and characteristics. These categories include carbohydrate, lipids, phenolic, terpenoids, and alkaloids, and other nitrogen-containing compounds [15]. Within each category, further division based on biogenesis or biosynthetic origin gives rise to different subcategories. Among these phytochemicals, antioxidant-rich phytochemicals in each category have been chosen to provide a further explanation of the biological activities, focusing on autophagy and antiaging effects.

In this review, six compounds were chosen from different categories of phytochemicals: three from polyphenols, one from terpenoids, one from alkaloids, and one from phytosterols (lipids). Polyphenols represent the largest category of phytochemicals and serve as powerful antioxidants due to their multiple hydroxyl groups [20]. Examples are carbonic acid, curculigoside, curcumin, rosmarinic acid, tectorigenin, tyrosine, etc. In this review, three compounds were chosen from polyphenols which are further classified as three categories: phenolic acids (CAPE), flavonoids (xanthohumol), and other phenolics (resveratrol). Furthermore, CDDO represents the terpenoids group, and guggulsterone falls into a phytosterol classification. Lastly, sulforaphane is an alkaloid phytochemical.

Due to an increase in the interest in natural products and phytochemicals in recent years, the role of natural compounds in modulating reactive oxidative species (ROS) and autophagy processes became a popular topic in scientific communities [21,22,23]. ROS refers to a number of oxygen-containing molecules and radicals that are produced through the oxidative metabolism within mitochondria or in response to external stimuli such as xenobiotics and bacterial invasion [24]. Of the major ROS, the superoxide anion (O_2_^•−^) usually serves as the starting point to form other free radicals, including hydrogen peroxide (H_2_O_2_) and the hydroxyl radical (HO^•^) [25]. In a recent review paper published by Thoma et al., they stated that the aberrant generation of ROS was directly related to the age-related loss of muscle mass, and effective targeted therapies can be developed [26]. The study included a few examples of vitamins and phytochemicals which target and regulate the ROS generation [27,28]. One of the examples is resveratrol; many studies demonstrated its capability of inhibiting lipid peroxidation and increasing catalase and superoxide dismutase activity [29,30]. Thus, it is worthy to investigate other phytochemicals’ antiaging effects through modulating the process of autophagy and the abnormal generation of reactive oxygen species.

The skin is the biggest organ of the human body and provides a physical barrier against exogenous factors such as ultraviolet radiation, microorganisms, and toxic substances [31]. The skin is composed of cells from several different systems of the organisms, such as the nervous, immune, circulatory, and endocrine systems. Evidently, skin senescence is closely related with organismal aging and age-related dysfunctions [32,33,34,35]. Therefore, understanding the impact of cell senescence on promoting systemic aging could result in new approaches to delay skin aging and age-related disorders. Skin aging is related to intrinsic factors and external agents; in both cases, the involved mechanisms are related to the malfunction of skin tissue cells, which lead to degradation of skin tissues. These changes come from the reduction in epidermal proliferation and extracellular matrix components synthesis, which are the result of imbalances of mechanisms explained above: the OSTA and homeostasis of autophagy.

In this paper, the role of reactive oxygen species and autophagic pathways related to the aging process will be thoroughly reviewed as well as introduce potential natural compounds that may promote antiaging effects by modulating reactive oxygen species and autophagy pathways. The effect of each compound on skin aging and skin health benefit will be briefly discussed, including in vitro and in vivo studies using different skin models. The compounds included in this paper are 2-cyano-3,12-dioxoolean-1,9-dien-28-oic acid (CDDO) derivatives (synthetic triterpenoids derived from naturally occurring oleanolic acid), caffeic acid phenethyl ester (CAPE, the active ingredient in honey bee propolis), xanthohumol (a prenylated flavonoid identified in the hops plant), guggulsterone (a plant steroid found in the resin of the guggul plant), resveratrol (a natural phenol abundantly found in grape), and sulforaphane (a sulfur-containing compound found in cruciferous vegetables).

## 2. Autophagy and Aging

Autophagy is the process by which eukaryotic cells selectively degrade and recycle misfolded, damaged, or excessive cytosolic products, including organelles and proteins [36]. While this process is present in basal conditions, autophagy is stimulated in the presence of stressors such as low nutrients or oxidized proteins. It is both a protective and adaptive mechanism that repurposes potentially toxic cytosolic proteins into necessary macromolecules that can be used to make energy [37]. Autophagy has three forms: microautophagy, macroautophagy, and chaperone-mediated autophagy (CMA) [38]. Because this process is degradative and changes in response to many stimuli, each step is tightly regulated.

Macroautophagy, the most common pathway of autophagy, is a multistep mechanism that begins with sequestering the organelles and proteins that will be sent for degradation. It is well-established that sequestering occurs through the formation and elongation of a double-membraned structure that may be referred to as an autophagosome or a phagophore. Once this de novo structure is formed, the autophagosome takes the products to be degraded to the lysosomes and fuses with it so that the lysosomal hydrolases can break down the contents [39,40]. The by-products of this process are formed into TCA-cycle intermediates that can be used to provide nutrients and generate energy. Each step of autophagy is tightly regulated by signaling proteins that respond to the amount of available nutrients. A lack of nutrients or starvation will initiate the process of autophagy.

There are several key protein complexes that initiate and regulate the phases of macroautophagy. Unc-51-like autophagy-activating kinase 1 (ULK1), a serine/threonine kinase and the mammalian functional homolog of yeast ATG1, is known to play a central role in initiating autophagosome formation during early autophagy [41,42]. ULK1 is a kinase complex with several protein partners, including ATG13, ATG101, and FIP200 (the focal adhesion kinase family interacting protein of 200 kDa) [43]. These proteins are targets of phosphorylation by AMP-activated protein kinase (AMPK) and the mammalian target of rapamycin (mTOR) [41,44]. When ATP is low, the nutrient sensor AMPK directly phosphorylates ULK1 which then initiates autophagy. This direct phosphorylation by AMPK is inhibited by mTOR when there are sufficient nutrients or in the presence of growth factors [45]. In essence, AMPK and mTOR oppositely regulate ULK depending on the availability of nutrients.

Other proteins that are a part of the autophagy apparatus have been used to provide insight into an organism’s ability to properly carry out autophagy within its cells. One example is the role of p62 in selective autophagy. In this case, p62 serves as a linking protein that directs ubiquitinated proteins to the autophagosome for degradation [46]. In order to carry out its function as an adapter, it first associates with LC3-II, a docking site for p62 bound to ubiquitinated substrates, within the growing phagophore. Once p62 has performed this task, it is later degraded within the lysosome. Because p62 levels should decrease with increased autophagy, the accumulation of p62 can serve as an indicator of poorly functioning autophagy machinery. The conversion of LC3 to LC3-II can also be evaluated in conjunction with p62 levels. Moreover, p62 has the ability to interact with major cellular control “switches” such as the nuclear factor-erythroid 2 p45-related factor 2 (Nrf2), caspases, and tumor necrosis factor (TNF)-α [47,48]. Taken together, depending on the levels of p62, the rate of autophagic turnover can influence other cellular mechanisms such as cell death, survival, and ROS regulation. Conversely, cellular stress and ROS accumulation may influence autophagy.

The other two forms of autophagy, microautophagy and CMA, also play a role in maintaining cellular homeostasis and function to complement macroautophagy. While maintaining the basic principle of proteomic recycling and degradation, microautophagy occurs through lysosomal invagination and the engulfment of cytosolic materials. Similar to macroautophagy, the process of microautophagy has been shown to be activated through nutrient deprivation and mTOR inhibitors such as rapamycin in mammalian tissues. The molecular mechanisms of microautophagy are not well-known in mammals; however, yeast studies have provided some insight. It has been shown that selective microautophagy utilizes hsc70 (the heat-shock cognate protein of 70 KDa), a cytosolic chaperone protein, in the early stages. CMA also utilizes cytosolic hsc70, which plays a role in targeting and directing the substrate proteins to the lysosome. Proteins are selected for degradation by the amino acid sequence, KFERQ, found within the proteins primary structure [49]. The proteins are then delivered to the cytosolic tail of the lysosome-associated membrane protein type 2A (LAMP-2A) receptor where the substrate protein is unfolded [50]. LAMP-2A then undergoes multimerization to allow translocation of the substrate protein into the lysosome for degradation. Because CMA and endosomal microautophagy are stimulated by stress or lack of nutrients, they often work in concert with macroautophagy to maintain the cells’ ability to constantly adapt.

As mammals age, the process of autophagy becomes less efficient and leads to the accumulation of potentially toxic cytosolic constituents. When the ability of cells to regulate potentially toxic levels of cytoplasmic proteins is diminished, it becomes easier for disease processes to emerge. Diseases in multiple systems have been related to decreased autophagy [51]. There is increased evidence that decreased autophagy leads to aging among different organisms. Studies that have been performed in model organisms such as yeast, nematodes, and fruit flies have shown that interfering with *Atg* genes and, by extension, macroautophagy significantly decreases lifespan [52,53,54]. There have been over 30 *Atg* (autophagy-related) genes discovered in yeast that are responsible for regulating this process [55]. Yeast *Atg* genes have served as a model for studying autophagy regulation in mammals. These *Atg* homologues are present throughout the early events of autophagy induction through association with *Beclin-1* and the recruitment of additional ATG proteins. In macroautophagy, ATG proteins play a key role in autophagosome formation and elongation through association with LC3 (microtubule-associated protein 1 light chain 3), a critical component in autophagosome formation [56]. Mice models have also been used to reveal the critical role that autophagy plays in lifespan. Selective deletion of *Atg5, Atg7, Atg16L1*, and *Beclin*, in mice, provided further insight into the importance of autophagy improving lifespan [57,58,59,60]. Changes in the negative regulator of autophagy, mTOR, have also been related to aging. As previously mentioned, mTOR is a nutrient sensor that is activated through phosphorylation when nutrients are high which, in turn, inhibits autophagy. It has been shown that the inhibition of mTOR promotes longevity in yeast and fruit flies. Aging organisms are associated with hyperphosphorylation of mTOR, which would lead to its hyperactivation, and inhibition of autophagy [61,62,63]. This would limit the cell’s ability to rid itself of oxidized proteins and maintain nutrient balance.

One of the hallmarks of aging is the progressive lack of function of normal cellular machinery, which makes a way for pathologies to emerge. There is mounting evidence that macroautophagy plays a critical role in maintaining normal neurological function. Because autophagy maintains protein turnover, it is necessary in order to prevent disorders such as Alzheimer’s disease and Parkinson’s disease, whose pathogenesis have been well-linked to the accumulation and aggregation of proteins [64]. Alzheimer’s disease is a form of dementia that is marked by the presence of senile plaques and intra-neuronal fibrillary tangles. These plaques are composed of toxic proteins and beta amyloid peptides. The reason these plaques accumulate is due to the hyperphosphorylation of Tau, a microtubule-associated protein, which stabilizes and therefore inhibits normal microtubule dynamics [65]. Because vesicular movement cannot occur without proper microtubule turnover, these plaques become trapped within the vesicles that are formed during the early events of autophagy and begin to accumulate. This further provides evidence that decreased autophagy plays a key role in the pathogenesis of this neurodegenerative disorder. Parkinson’s disease is a devastating movement disorder that gets progressively worse over time. Its pathogenesis is due to neuronal cell death in the substantia nigra and accumulation of alpha-synuclein. It has been shown that the degeneration of autophagy (both macroautophagy and CMA) plays a role in the development of Parkinson’s disease [66,67]. There have been studies highlighting the decreased expression of major CMA components LAMP2A in patients with this disease. There have been many studies confirming that CMA markedly decreases with age, primarily through the decreased presence of functional LAMP2A. With a better understanding of the interplay between autophagy and aging, we can develop therapeutics to target these molecular changes.

Lastly, it is important to note that apoptosis and autophagy are closely related molecular processes, which maintain organismal and cellular homeostasis. Whereas autophagy maintains cellular homeostasis by recycling selective intracellular organelles and molecules, apoptosis fulfills its role through dismantling damaged or unwanted cells. In this review, we highlighted the antiaging effects of those natural compounds that demonstrated organismal and cellular homeostasis through autophagy and apoptosis.

Two distinct pathways are responsible for the induction of apoptosis: intrinsic and extrinsic pathways. The release of cytochrome c into the cytosol activates the intrinsic pathway, which leads to cytochrome c binding to and forming a complex with Apaf-1, causing caspase-9 to be activated. The activation of caspase-9, in turn, activates caspase-3. The extrinsic pathway begins with stimulation of the TNF family of death receptors which leads to the activation of caspase-8, resulting in the direct activation of caspase-3 [68]. It is worth to note that the compounds that are included in this review utilize both autophagy and apoptosis pathways to maintain organismal and cellular homeostasis to promote longevity.

## 3. Oxidative Stress and Aging

ROS are molecules constantly being generated through both cellular metabolism in the mitochondria and exposure to various environmental stimuli [69]. ROS include both radical (superoxide anion and hydroxyl radical) and nonradical species of oxygen (hydrogen peroxide). “Free radicals,” which refers to atoms with unpaired electrons, are highly reactive and can cause oxidative damage in tissues. During cellular respiration, the inefficient transfer of electrons between molecules allows for some electrons to slip through and bind with oxygen, thus forming these “free radicals” [70]. Once a radical has been produced, it can be converted to a non-radical form, or it can interact with other components of the cell and cause damage. Because of the deleterious effects of ROS accumulation, the body has both enzymatic and non-enzymatic antioxidant systems in place to prevent ROS from rising to a harmful level. When the oxidant and antioxidant systems are out of balance, oxidative stress that may damage the cells is generated [71]. The damage produced by oxidative stress has been implicated not only in many human diseases but in the process of aging as well.

As the key player in metabolism, mitochondria are the major producers of reactive oxygen species [72]. Given the proximity of mitochondrial DNA to ROS production, some studies suggest that mitochondrial DNA is more susceptible than other genetic materials to ROS-induced damage and mutations [73]. Mitochondrial DNA encodes for proteins used for the regulation of critical processes such as oxidative phosphorylation and apoptosis. It is inferred that increased mutations in these regions may lead to the malfunctioning of these processes, and thus the generation of more ROS that ultimately leads to cell death [74]. Studies have shown that dysfunctional mitochondria and increased production of oxidative stress play a large role in the pathophysiology of neurodegenerative diseases, including Alzheimer’s disease, Parkinson’s disease, and Huntington’s disease [75]. Overall, aging and age-related diseases are strongly related to mitochondrial decline.

As mentioned before, the accumulation of free radicals and ROS has the tendency to cause oxidative damage to organelles and other macromolecules such as proteins, lipids, and DNA—this has been termed the oxidative stress theory of aging [71,76]. It has been proposed that the overproduction of ROS promotes cellular senescence, a state that occurs as a response to cellular damage and stops proliferation. The increased ROS accumulation is reported to enhance telomere shortening which has the most correlation with cellular senescence [77]. This senescent state is associated with a number of secretory proteins that play roles in age-related disease. Inducing senescence in stem cells that are necessary for the regeneration and repair of damaged tissues can impede an organism’s ability to maintain homeostasis and promote longevity. It has been well-documented that with age, stem cells lose their ability to regenerate [78,79,80,81]. In addition, studies using animal models demonstrate that increased ROS production leads to a decline in stem cell renewal. In one study using a mouse model, researchers deleted the Ataxia telagiectasia-mutated (ATM) kinase gene in hemopoietic stem cells (HSC) [82]. The deletion of the ATM kinase resulted in increased ROS and a decreased ability for cellular renewal. However, the oxidative stress was ameliorated using antioxidant N-acetylcysteine and led to regaining function of those cells. This suggests that the accumulation of ROS could be the main factor that caused the decline in the HSC function.

The generation of ROS can also interact with a number of transcription factors and regulatory proteins. A key regulatory factor of redox balance is the Nrf2-Kelch-like ECH-associated protein 1 (Keap1) system, which upregulates genes that are responsible for antioxidant functions [83]. The transcription factor, Nrf2, binds to the promoter region of antioxidant response elements which is involved in cytoprotective mechanisms, including DNA repair, proteostasis, and the prevention of apoptosis. When Nrf2 functions are low, ROS accumulation is unchecked, and cells become more sensitive to oxidative stress [84]. The decreased function of Nrf2 has been shown to promote molecular mechanisms that are representative of aging, such as telomere shortening, genomic instability, mitochondrial dysfunction, and cellular senescence [85,86,87,88].

Telomerase reverse transcriptase (TERT) is a protein associated with promoting telomere extension. Recently, it has been shown that increased Nrf2 expression is associated with increased TERT expression, and thus continuous cellular division [85]. As reported before, Nrf2 was reported to prevent genomic instability through promoting DNA repair [86,87]. With age, DNA repair mechanisms such as replication errors and base excision repair occur at a much higher rate which leads to a number of pathological conditions [89]. As previously mentioned, mitochondrial accumulation of ROS and mitochondrial dysfunction increased with aging. As an antioxidant-promoting transcription factor, Nrf2 plays an important role in maintaining redox balance within the mitochondria. It has also been shown that downregulation of Nrf2 could be associated with cellular senescence. In a study using fibroblasts, Kasai et al. suggested that younger cells were less sensitive to oxidative stress when compared with the older ones [90]. When treated with a Nrf2 activator, the older cells were less sensitive to oxidative stress. Taken together, it is well-established that Nrf2 is an important player maintaining oxidative balance, homeostasis, and thus longevity.

The production of ROS is a continuous process that becomes less efficient with age. The presence of ROS has multifaceted effects on cellular function and generation of disease. Transcription factors such as Nrf2 play a key role in maintaining antioxidant function to keep oxidative stress at a minimum. When master regulators like Nrf2 decline in activity with age, the organisms are more sensitive to oxidative stress, which leads to aging-related pathophysiological conditions and disorders. Understanding the important role ROS play in aging leads to antioxidants as a potential remedy in the prevention and treatment of aging-related diseases.

## 4. Antiaging Compounds

Compounds derived from various plant sources have been used widely around the world for many centuries as an alternative remedy for chronic and aging-related diseases. In this section, we evaluated the action and mechanism of several phytochemicals on oxidative stress, autophagy, and other aging-related conditions. These phytochemicals include CDDO derivatives, caffeic acid phenethyl ester, xanthohumol, guggulsterone, resveratrol, and sulforaphane with their structures presented in Figure 1.

### 4.1. CDDO Derivatives

Triterpenoids are naturally occurring compounds that can be found in the wax-like covering of fruits, such as apples, figs, cranberries, and olives, and herbs, such as lavender, oregano, thyme, and rosemary. These compounds, which are formed through the cyclization of squalene, resemble steroids in structure and have been used for centuries due to their antibacterial, anti-fungal and anti-inflammatory properties [91]. Oleanolic acid, which is primarily extracted from olive leaves, is a pentacyclic triterpenoid that has been shown to display weak anti-inflammatory effects. In an attempt to strengthen these effects, synthetic derivatives were created more than 20 years ago, including 2-cyano-3,12-dioxooleana-1,9(11)-dien-28-oic acid (CDDO) and its C28-modified derivative, methyl-ester (CDDO-Me, also known as bardoxolone methyl) [92]. CDDO and CDDO-Me have been shown to display potent anticancer effects through the inhibition of processes that promote proliferation, angiogenesis, and metastasis.

As mentioned above, there are three major mechanisms of autophagy: macroautophagy, microautophagy, and chaperone-mediated autophagy (CMA), which involve the delivery of substrates to the lysosome for degradation. Recent studies have identified CDDO derivatives may inhibit necroptosis by promoting the process of CMA via inhibition of heat shock protein 90 (Hsp90).

A number of recent studies have identified key mechanisms that moderate the potent effects of triterpenoids. The therapeutic effect of CDDO comes from its upregulation of the master antioxidant transcription factor Nrf2 through the conformational change of Nrf2-inactivating Keap1. This mechanism involves the upregulation of phase II antioxidant genes through the activation of Nrf2-dependent transcription, thereby elevating intracellular antioxidant defense systems to battle oxidative stress. Triterpenoids also directly inhibit nuclear factor-κB (NF-κB) signaling, which is a key pathway that plays a role in regulating the production of many inflammatory mediators and their signaling cascades: TNF-α, IL-1β, INF-γ, and TLR. Inflammatory cells synthesize ROS and nitrogen, which creates oxidative stress in nearby cells, causing a recruitment of more inflammatory cells to their location [93]. A rise in the level of antioxidant enzymes produced by Nrf2 reduces the cellular levels of ROS, which in turn induces NF-κB signaling and the transcription of pro-inflammatory genes such as TNF-α and inducible nitric oxide synthase (iNOS). The ability of CDDO-Me to limit the synthesis and secretion of neurotoxic pro-inflammatory cytokines and inducing ROS accumulation suggests that long-term administration of these brain-permeable synthetic triterpenoids may confer neuroprotection in vivo [94].

In addition to the anti-inflammatory properties of CDDO, synthetic oleanane triterpenoids (SO) may possess cytoprotection properties by inactivating Keap1. SO activates the “phage 2” response, an intrinsic mechanism used by cells to deactivate electrophilic or oxidative stress [95]. Both in vitro and in vivo studies showed that SO are among the most potent known inducers of the phase 2 [96,97,98,99]. Liby et al. demonstrated that nanomolar concentrations of SO increased the expression of the cytoprotective heme oxygenase-1 enzyme in cell culture assays. Moreover, they demonstrated that treatment with one of the SOs caused the elevation of protein levels of Nrf2 by increasing the expression of a number of antioxidant and detoxification genes regulated by Nrf2. Likewise, when mice were injected with CDDO-Im (imidazole) intraperitoneally, and the expression of iNOS protein and the production of nitric oxide (NO) were measured, the group observed that synthetic triterpenoids (CDDO-Im) suppressed the activation of macrophages by decreasing the production of NO, suggesting cytoprotective properties of SO [100].

Initially, CDDO was developed as a highly potent anti-inflammatory agent based on the structure of oleanolic acid. The poor oral bioavailability of CDDO ultimately led to many other derivatives being synthesized, including CDDO-Me (methyl ester) and CDDO-Im [100,101]. The health benefits of these molecules are getting more attention from the science community due to anti-inflammatory, antioxidant, and anti-angiogenic properties [102,103,104,105].

Both in vivo and in vitro studies demonstrated multiple pathways in the regulation of angiogenesis [96,100,106]. Currently, there are three known mechanisms of action of CDDO-Me and CDDO-Im that relate to the regulation of angiogenesis. The first target is the suppression of NF-κB signaling, which leads to the suppression of angiogenesis. Both CDDO-Me and CDDO-Im have been shown to possess potent inhibition activities of IκB kinase, which activates NF-κB. The second target is the suppression of the Signal Transducer and Activator of Transcription (STAT) signaling, as mediated by the ability of both CDDO-Me and CDDO-Im to suppress the phosphorylation of both STAT3 and STAT5, which is required for the activity of these proteins as transcription factors. STATs are known to play an important role in mediating both wound healing and angiogenesis. A third important pathway regulated by synthetic oleanane triterpenoids is transforming growth factor-h/BMP/Smad signaling which, in turn, allows the triterpenoids to enhance their transcriptional activity of the respective Smads that are involved in regulating gene activity. There is plenty of evidence for the role of this transforming growth factor signaling as a critical regulator of both wound healing and angiogenesis. A fourth molecular target of CDDO-Me is Keap1, the endogenous inhibitor of the transcription factor Nrf2, although the effects of genes regulated by Nrf2 on the process of angiogenesis is still unclear [106].

The skin is one of the organs that shows obvious and visible signs of aging when one becomes older. Therefore, promoting antiaging effects on the skin has earned tremendous interest in recent decades. Physiological processes and molecular pathways that induce cutaneous aging are related to other cellular pathways. For example, a decrease in the autophagy process within the basal cell layer results in a reduction in the proliferative ability of cells, including keratinocytes, fibroblasts, and melanocytes [107]. Moreover, ROS play a critical role in skin aging by inhibiting the activity of protein tyrosine phosphatases and triggering downstream signaling pathways, including the activation of mitogen-activated protein kinase (MAPK) and subsequent NF-κB and transcription factor activator protein-1 (AP-1) [108,109,110]. In addition, senescence of skin keratinocytes are increased due to oxidative stress caused by the elevation of LC3-II and p63 protein levels [111]. Many in vitro and in vivo studies showed that oleanolic acid derivatives are powerful inhibitors for oxidative stress caused by the induction of NO synthase and cyclooxygenase 2 [110,112,113]. Choi et al. used a murine model to show CDDO derivatives decreased skin lesions and the infiltration of immune cells caused by ROS. Kim et al. also demonstrated that CDDO derivatives decreased the autophagy process in keratinocytes by evaluating the protein expression of LCE-II and p62 by Western blotting [113].

Evidence from the collected data suggested that CDDO and CDDO derivatives actively modulate the bioactive molecules that are responsible in the pathogenesis of antiaging processes such as autophagy and oxidative stress. A major molecular pathway that CDDO promotes the initiation of the autophagy process is upregulating the transcription factor Nrf2 and transcription factor EB (TFEB). Moreover, it promotes longevity by modulating many inflammatory mediators such as Keap1 and iNOS. Based on this evidence, clinical trials are needed to explore CDDO and its derivatives effects for the purpose of future therapeutic modalities.

### 4.2. Caffeic Acid Phenethyl Ester (CAPE)

Caffeic acid phenethyl ester (CAPE) is a naturally occurring compound with many known healing effects on the human body. The full chemical of CAPE is 2-phenylethyl (2E)-3-(3,4-dihydroxyphenyl) acrylate. It can be extracted and purified from the beehive propolis, which is a substance collected from various plant sources by honeybees [114]. Although CAPE is a naturally occurring hydrophobic polyphenol, it can also be prepared through enzymatic synthesis [115,116,117,118]. CAPE has been shown to provide numerous beneficial effects, in both in vitro and in vivo studies, through its ability to function as an anti-inflammatory, antioxidant, and an anti-tumor agent. The bioactivity of CAPE is largely attributed to the presence of the catechol ring and hydroxyl groups within its structure [119,120]. Through structural modifications of the catechol ring, several studies have highlighted its importance to CAPE’s overall function and activity.

To add to the multifaceted beneficial effects of CAPE, there have been studies linking this propolis extract to mechanisms of autophagy. The nature of CAPE’s role in autophagy may be dependent on the physiological state of the cell. There have been several cancer studies highlighting CAPE and its derivatives to upregulate autophagy activity. In the human neuroblastoma cell line SH-SY5Y, treatment with CAPE was shown to induce autophagy activity [121]. Through Western blot, this study found increased expression of the LC3-II protein, an important constituent of the autophagosome structure, after treatment with CAPE for 4, 8, and 12 h. Similarly, a study conducted using breast cancer cell lines MDA-MB-231 found that CAPE caused an increase in LC3-I to LC3-II conversion and upregulation of LC3-II in cells that were LPS-stimulated [122]. CAPE has also been shown to induce autophagy in a liver fibrosis study which utilized both in vivo and in vitro models. In this study, CAPE ameliorated the CCl4-induced liver fibrosis by inhibiting the TGF-β/Smad pathway. Additionally, CAPE induced autophagy in HSC cell lines and thereby prevented fibrosis. Evidence of CAPE’s autophagy activation was measured in HSC-T6 cells through immunofluorescent visualization of autophagosome formation and Western blot analysis of autophagy markers such as LC3II, ATG7 and BECLIN 1. Because there was decreased expression of pAKT and pmTOR, it was suggested that CAPE’s induction of autophagy was through the inhibition of AKT/mTOR signaling.

Other derivatives of CAPE have been observed to alter autophagy mechanisms in cancer cell lines. One such derivative, phenethyl caffeate benzoxanthene lignan (PCBL), was shown to increase autophagy in WiDr colorectal adenocarcinoma cells [123]. PCBL also increased both BECLIN 1 expression in Western blot analysis and autophagic flux in WiDr cells. Further experimentation suggested that PCBL induced autophagy through a class III PI3-kinase-dependent manner, also known as the canonical pathway, rather than through mTOR regulation. In addition, increased autophagic activity in glioma cells through increased expression of LC3-II and decreased p62 expression was observed after PCBL treatment [123].

Many studies have shown that phytochemicals have the potential to offer antiaging properties through scavenging reactive oxygen species, enhancing production of antioxidants, and maintaining the cytosolic balance of proteins [124]. In a study using *Caenorhabditis elegans,* CAPE was shown to promote longevity and resistance to stress through the modulating insulin-like DAF-16 pathway, a mechanism responsible for regulating oxidative stress in nematodes [125]. In this study, CAPE significantly increased DAF-16 translocation into the nucleus which indicates increased activity of the transcription factor. DAF-16 also appeared to play a role in promoting longevity, as lifespan was significantly decreased in the DAF-16 loss-of-function mutant *C. elegans.* After treatment with CAPE, the accumulation of ROS was significantly decreased. Similarly, CAPE reduced ROS, increased lifespan, and increased activity of the transcription factor, Nrf2, in human Hct116 cells.

There is a plethora of evidence that ROS accumulation promotes aging [74,76]. To investigate the role of antioxidants in the amelioration of ROS-induced aging, CAPE and melatonin were used in a mouse model to observe its effects on cardiovascular aging [126]. After CAPE and melatonin were administered to the aged mice, the heart tissues were evaluated through histological observations, antioxidant enzyme assays, and protein determination through absorbance measurements. The authors found that CAPE increased expression of antioxidant enzymes and reduced age-related ultrastructure modifications in the heart and aorta. This study concluded that CAPE could play a role in reversing and delaying the aging process in the cardiovascular system.

Antiaging research related to ultraviolet (UV) rays and how they can damage skin have received remarkable spotlights. Solar UV exposure is a major causative factor for age-related changes of skin, such as wrinkles and cancer development. In other words, UV radiation can induce skin pathologies, including erythema and inflammation, degenerative aging change, and cancer [127]. UV radiation produces both direct and indirect DNA damage, which can result in mutagenesis in skin cells [128]. In recent studies, CAPE was found to possess various activities such as antimicrobial, antioxidant, anti-inflammatory, and cytotoxicity activities. These activities were demonstrated by evaluating CAPE’s ability to suppress ROS levels in different cells and tissues [129,130,131,132,133]. Shin et al. demonstrated that CAPE could block UV-stimulated matrix metalloproteinase-1 (MMP-1) levels in skin cells and in human skin tissues by inhibiting histone acetyltransferase activity, which may lead to the attenuation of non-histone proteins and histone H3K9 acetylation. Therefore, CAPE may prevent skin aging caused by UV exposure [128]. Extensive literature is available on the antiaging effects of CAPE, but still some areas require to be explored, such as pre-clinical and clinical studies. Further investigation to use this data of CAPE’s bioactive activities to become a therapeutic treatment of ailments is required.

### 4.3. Xanthohumol

Hops flowers (*Humulus lupulus* L.) have been used widely in the brewing of beer to give it the characteristic bitter flavor and aroma. Besides being used in beer brewing, hops flowers contain a bioactive substance called xanthohumol (XN). XN is found in the trichomes on the hop leaves and secreted as part of the hop resin, with a content of 0.1–1% dry weight [134]. The traditional extraction of XN was performed by repeated chromatographic steps on silica gel using different solvents. This has been converted to a more time-efficient isolation and purification method by developing a high-speed counter-current chromatography. XN, which is a prenylated flavonoid derived from the female inflorescences of the hops, has been used for its diverse pharmacological properties which may include: anticancer, antiaging, antibacterial, anti-inflammatory activities, etc. [135].

XN has been extensively researched and has shown efficacy in regulating several pathways related to proliferation, apoptosis, as well as having the ability to inhibit NF-κB and Akt (protein kinase B) activation in vascular endothelial cells. Some studies have shown that XN contributes to the prevention of oxidative damage by reducing ROS and NO production. Another pathway that it acts upon is the inhibition of cyclooxygenase activity which leads to anti-inflammatory effects [136]. XN is able to induce cell-cycle arrest leading to the halt of mitosis. The most important mechanism of inhibiting apoptosis by XN consists of modulation of the expression of different pro-apoptotic factors, such as caspases-3, -7, -8, and -9, and downregulating the expression of apoptosis regulator gene B-cell lymphoma 2 (*Bcl-2* ) [137].

XN is shown to have possible protective effects on brain disorders secondary to aging [138]. In clinical trials, XN was observed to protect against the excessive production of pro-inflammatory cytokines IL-1β and TNF-α and provided a balance between pro- and anti-apoptotic factors in the brain, in a dose-dependent manner. In addition, XN was noted to induce an increase in the expression of brain-derived neurotrophic factor, which may lead to modulation of the inflammatory response in the brain. Aging increases the protein and mRNA expression of the glial fibrillary acidic protein, an astroglia marker in the brain. In recent trials, the administration of XN to mice showed a significant decrease in these aging-related markers, suggesting that XN may have possible protective factors against degenerative diseases [138].

As a nonspecific protein degradation pathway, autophagy is designed to control the progression of cancer and neurodegenerative diseases. Recent research indicated that autophagosome maturation of human epidermoid carcinoma A431 cells is restricted by XN. This impairment is caused by the binding of XN to the *N*-domain of the valosin-containing protein (VCP), which inhibits the action of VCP. VCP is an essential protein for autophagosome maturation [139]. The modulation of autophagy by XN is the main contributor to the mechanisms underlying the anticancer activity of XN, although further studies are needed to illustrate whether this autophagy inhibits or facilitates the XN-induced apoptosis [134]. Angiogenesis is essential for tumor growth in the growing stages of cancer and is characterized by the growth of new blood vessels to travel farther within the body. Angiogenic factors include MMP and vascular endothelial growth factor (VEGF). Some studies that focus on the antiangiogenic effects of XN portray that it has the ability to inhibit important pathways by suppressing enzymes involved in the proliferation, endothelial cell migration, and ultimately the angiogenesis of human microvascular endothelial cells in a dose-dependent manner [137].

XN showed antioxidant activities by directly reducing ROS and indirectly inducing the cellular defense mechanisms to overcome the oxidant stress [140,141,142]. Moreover, the pharmacological targets for its bioactivity have been identified and investigated, including Keap1 and NF-κB, as well as binding to the VCP and MRP5 [143,144]. Among these targets, Keap 1 is an important target that is responsible for regulating autophagy and the antiaging process [143,145].

In skin aging, there is degeneration of the extracellular matrix’s collagen and elastin fibers, and from its reduced biosynthesis and increased degradation by elastase and MMPs. Philips et al. evaluated XN’s ability to inhibit MMP-1, MMP-8, and elastase activity and to stimulate biosynthesis of fibrillar collagens, elastin, and fibrillins in dermal fibroblasts [146]. XN was shown to prevent the loss of proteoglycan and collagen as well as the MMP-dependent shedding of I-CAM in skin cell culture medium. These activities suggested a role in reversing skin aging [147].

Although there are many in vivo and in vitro studies to demonstrating XN’s beneficial effects on human health, developing XN as a reliable drug can be challenging [148]. Extracting and isolating the pharmacological-relevant concentrations from food sources are difficult due to its low solubility in water [135]. Similar to other phytochemical products, further studies need to be performed to develop XN as a reliable drug for specific therapeutic applications.

### 4.4. Guggulsterone

Guggulsterone (GS) is a yellow resin isolated from a small thorny tree called Commiphora mukul. Widely used in Asian countries such as India, GS has traditionally served as herbal medicine for inflammatory diseases for centuries [149]. GS is a plant sterol, and its hypolipidemic effects have been well-studied in in vitro and in vivo animal models [150] [151]. Chander et al. reported cardioprotective effects of GS mediated through the inhibition of ROS production in rodents [152]. Because ROS is one of the fundamental causes of oxidative stress, which is implicated in the aging process, GS’s capability in decreasing ROS will serve as a strength against aging. In this study, the authors synthesized a synthetic guggulsterone (E and Z-isomers) and tested in human low-density lipoprotein and rat microsomes in comparison to natural GS and gemfibrozil. The study results suggested that GS inhibited oxidative degradation of lipids by counteracting against the generation of superoxide anions and hydroxyl radicals [152]. Additionally, GS was found to decrease cholesterol through antagonizing the farnesoid X receptor (FXR) [153]. Interestingly, GS increased the expression of FXR in 3T3-L1 adipocytes, suggesting that it may act as an inverse agonist at low doses [154]. Likewise, in the same adipocyte model, GS also increased the expression of transcriptional co-activator PGC1α [155], a key regulator of mitochondrial biogenesis and a potential therapeutic target for age-related disorders [156].

GS serves as an antagonist for the FXR, an important nuclear receptor that regulates an array of genes, such as bile acid expression, lipids, and, most importantly in our investigation, autophagy [157]. However, to greatly understand the role of GS as an antagonist of FXR in increasing autophagy, it was hypothesized that GS will be able to increase autophagy through the same mechanism in which it decreases cholesterol. Another study noted that GS decreased cholesterol through antagonism of the FXR receptor, which increased the expression of the bile salt export pump [158]. The bile salt export pump is a transporter that is mainly responsible for the excretion of bile salts into bile, which decreases the synthesis of cholesterol in the liver. GS’s ability to increase the expression of the bile salt export pump through the antagonism of the FXR shows that the compound was able to decrease cholesterol and demonstrate hypolipidemic effects. Similarly, autophagy is indirectly regulated by FXR, in which the activation of FXR suppresses the process of autophagy in the liver. In other words, the process of autophagy was increased through the antagonisms of the FXR. Furthermore, skin health benefits of guggulsterone were discovered in in vitro and in vivo studies [159,160]. Sarfaraz et al. discovered that the topical application of GS to SENCAR mice resulted in a significant decrease in skin edema, hyperplasia, and classical markers of inflammation and tumor promotion, such as COX-2 and iNOS. They found that the topical application of GS induced inhibition of phosphorylation of MAPKs, activation of NF-κB/p65 and IKKα/β, and degradation and phosphorylation of IκBα [160]. In addition, Koo et al. showed that GS suppressed tyrosinase activity in B16 murine melanoma cells and melanin biosynthesis which promotes longevity of melanin cells [159].

Despite the differences in the results across various studies, the collective data suggest that GS has an inherent ability to produce beneficial antiaging effects. Although GS’s potential is vast and promising, more studies and research are needed to investigate its role in aging. While there are many studies regarding its effects on the conditions and therapeutic effects that can indirectly be linked to the aging process, more extensive studies need to be completed to investigate its direct effects on the aging process.

### 4.5. Resveratrol

Resveratrol is a natural polyphenol found in red wine and plants, such as plums, grapes, and raspberries, and has been well-studied and found to work in multiple mechanisms and have a plethora of benefits [161]. One of the mechanisms by which resveratrol works is through the facilitation of oxidative stress and mitochondrial dysfunction [161]. As previously discussed, these factors have been shown to contribute to the aging process through the free-radical and mitochondrial theories of aging [162,163]. A study that combined resveratrol and exercise in old mice reported evidence of improved antioxidant activity and decreased lipid peroxidation in the major organs [164]. The study noted the varying effects on the organs and a need to design a better intervention, but the data suggested that resveratrol and exercise is beneficial in the prevention of age-related diseases. Another study evaluated aging secondary to oxidative stress in rat cardiomyocytes after treatment with resveratrol. In this study, treatment with resveratrol decreased the concentration of NO in cardiomyocytes from 18.49% in 2 months to 62.67% in 8 months [165]. In addition, treatment with resveratrol caused a significant decrease in nitric oxide oxidative stress markers and total lipoperoxidation during the aging process, thus concluding that the administration of resveratrol during the aging process may help decrease oxidative stress.

Because resveratrol is a known antioxidant, it is no surprise that this compound has been shown to produce protective effects by attenuating oxidative injury. One study highlighted that resveratrol is able to accomplish its protective effects through the activation of TFEB which is a regulator of autophagy [166]. Human umbilical vein endothelial cells (HUVECs) treated with resveratrol demonstrated increased TFEB activity. Furthermore, the study concluded that resveratrol offered protective effects to HUVECs from oxidative damage by modulating autophagy in a TFEB-dependent manner. Resveratrol was shown to modulate the acetylation state of proteins to regulate autophagy and promote longevity [167,168,169]. The lifespan in C. elegans and mice were prolonged by resveratrol via activation of the NAD^+^-dependent deacetylase SIRT1. In these studies, researchers showed that deletion and depletion of essential autophagy genes, such as bec-1 in C. elegans and Atg7 in yeast and flies, resulted in a decreased lifespan of the organisms.

Natural compounds are continuously being studied because they are linked to many health benefits. Resveratrol is a powerful antioxidant, and in recent studies, resveratrol was studied in cosmetics, specifically its effect on human skin lightening [162]. As skin ages, there is a progressive decline in the function compared to healthy skin [170]. Healthy skin appearance is expanding the cosmetics industries as the consumers have demanded more skin care products that can maintain healthy-looking skin. In the study of resveratrol effects on skin, it was observed that resveratrol can whiten human skin and slow down aging by a multitude of mechanisms [162]. Mechanisms include the inhibition of catalyzation of human tyrosine, suppression of gene maturation of tyrosine, inhibition of ROS, enhancement of cellular antioxidant capacity through Nrf2 mechanisms, attenuation of inflammatory responses, and inhibition of MMP’s catalytic activity. Several studies were conducted to evaluate the effects of resveratrol in humans with various health conditions. While a number of these studies demonstrated a decrease in oxidative stress with resveratrol supplementation [171,172,173], others showed no effects [174,175]. Differences in experimental design, including sample size and dose of resveratrol, might have contributed to the contrasting results seen in various studies. Overall, clinical trials have shown that resveratrol is well-tolerated and safe with very few reports of adverse effects [176,177,178].

### 4.6. Sulforaphane

Sulforaphane (SFN) is an isothiocyanate found in cruciferous vegetables, such as broccoli and Brussel’s sprouts [179,180,181]. SFN has been well-studied as a chemo preventive agent, as previous epidemiological studies indicate that dietary intake of these vegetables may protect against cancer. In a study on SFN on Caco-2 cells with rapamycin, it was highlighted that the chemo-preventative effects were enhanced by multiple mechanisms, including Nrf2, human pregnane X receptor -mediated UGT1A1, UGT1A8, and UGT1A10 induction [182]. The results of the study indicated that during cancer initiation, SFN may augment the detoxification and removal of carcinogens by regulating multiple signaling pathways to suppress cell growth and induce cell death.

SFN majorly works by inducing the Nrf2 pathway and inhibiting NF-κB, thus activating the antioxidant and anti-inflammatory responses. In one study, SFN pre-treatment was shown to attenuate oxidative stress and pro-inflammatory response, and thereby modulate post-ischemic ventricular function in isolated hearts submitted to ischemia–reperfusion [183]. A cardio-protective effect was found in another study in rats given broccoli. The study was over a period of 30 days, with rats in isolated heart preparations that resulted in reduced apoptosis and increased activity of antioxidant enzymes [184].

SFN’s ability to induce autophagy was studied using transmission electron microscopy on sulforaphane-treated PC-3 (adenocarcinoma cell line) cells. SFN-treated cells displayed key features of cells undergoing apoptosis, such as the formation of membranous vacuoles and chromatin condensation in the nuclei [185]. This study reports that it is reasonable to assume that SFN-induced apoptosis in human prostate cancer cells was derived from an increase in autophagy activity as determined by their analysis of sub-diploid cells by flow cytometry, following staining with propidium iodide [185]. The histograms resulted with SFN treatment showed a 6-fold increase in apoptotic cells.

Although several studies show that SFN is able to reactivate the Nrf2 pathway, it was only when enough SFN was in the tissues that the process activates. The concentrations of SFN used were also taken into account with the various studies because different SFN concentrations have different effects, depending on the cell type of the organism. In particular, one study with red flour beetles reported that a diet supplemented with 1% *w*/*w* broccoli concentration increased the beetle’s longevity in conditions of external stressors [184]. Transcription factor Nrf2 induced detoxification then led to increased stress resistance for the flour beetles. In a phase II clinical trial, men with recurrent prostate cancer were supplemented with a single daily dose of 200 μmol of SFN extracted from broccoli sprouts for a period of 20 weeks. The dose was chosen because prior studies showed this dose led to low micromolar intro-prostatic concentrations, displayed tolerability and safety with similar doses of these extracts, and it was not sensible to treat with higher doses of these extracts [184]. Although the goal of a greater than 50% PSA decrease was not met in this study, SFN still showed effectiveness and is still being studied in additional trials. More pre-clinical studies are needed in order to determine the most effective dosing, but the two studies we discuss here show promising potential for SFN to be used in pharmacotherapy.

More recent studies have investigated SFN’s antiaging effects and have discovered that its mechanism is through preservation of proteostasis. This mechanism activates the proteasome subunit PSMB5 through the upregulation of heat-shock proteins, specifically Hsp27, which leads to increased cellular lifespan and prevention of neurodegeneration [186]. In a microarray analysis, it was shown that proteasome activity was attributed to the SFN-mediated upregulation of Hsp27 [181]. Treatment with SFN enhanced the expression and function of proteasome against oxidative stress. As SFN activates the heat-shock proteins in the bodies, the proteins assist with slowing down the aging process. The gradual deterioration in cells are usually induced by oxidative stress, and lately, there are studies on SFN against cellular skin aging [187].

There are currently more studies going on to use SFN as a natural ingredient for antiaging creams. Skin is sensitive against multiple external and internal factors, and over time, its function and appearance deteriorate. SFN has also demonstrated protective effects against ultraviolet-induced skin damage through several mechanisms of action and the maintenance of collagen levels during photo-aging via the inhibition of the AP-1 activation and expression of metalloproteinases. Cell maturity is related to the progressive rate of glucose metabolism through glycolysis and this is countered by SFN treatment [187]. Glycolytic restriction was previously found to delay senescence, and a study on SFN on fibroblast senescence shows evidence of restricted cellular glucose uptake and increased glycolysis. Overall, experimental findings show evidence of the benefits to SFN, but more studies need to be investigated and performed to fully gauge the potential of SFN.

## 5. Conclusions

Evidence from different in vitro and in vivo studies demonstrated that there is a positive correlation between the progress of aging and the decline in autophagy and elevation of ROS. In other words, an impairment in autophagy or an imbalance of ROS induces pathological aging and disease. Studies have shown that long-term health benefits and antiaging effects may be achieved by promoting autophagy and suppressing oxidative stress in various tissues and organisms.

Promising natural compounds that demonstrated antiaging effects through a modulating autophagy process and reducing ROS are reviewed in this article. The majority of these chemicals demonstrated pharmacological effects via multiple molecular pathways. To briefly summarize the mechanisms of each compound, the compounds are categorized based on their mechanism of actions. CDDO and resveratrol upregulated the transcription factor Nrf2 and TFEB, respectively, which caused the inhibition of mTOR pathways to promote the initiation of the autophagy process. SFN and CAPE not only demonstrated promoting the autophagy process by inducing Nrf2 and LC3 compounds but also showed that these two compounds demonstrated a reduction in ROS in various organisms via inhibiting NF-κB and upregulating the expression of autophagy-related genes such as *Atg 7* and *Beclin 1* (CAPE) and Hsp27 (SFN). The major antiaging mechanism for xanthohumol and guggulsterone was demonstrated by inhibiting NF-κB pathways and resulted in anti-inflammatory effects and anti- oxidative stress in various organisms. The antiaging mechanisms of those phytochemicals through regulating autophagy and oxidative stress are summarized in Figure 2 and Figure 3.

Many pharmacological drugs have been shown to induce changes in autophagy and ROS, contributing to their health benefit; however, natural compounds are continuously being investigated because they are linked to other health benefits. Although overwhelming evidence in the literature supports that the discussed natural compounds have the potential for antiaging effects, further studies, including pre-clinical and clinical, need to be performed to understand and identify specific drug targets and doses for therapeutic intervention in antiaging effects.

## Figures and Tables

**Figure 1 molecules-27-04396-f001:**
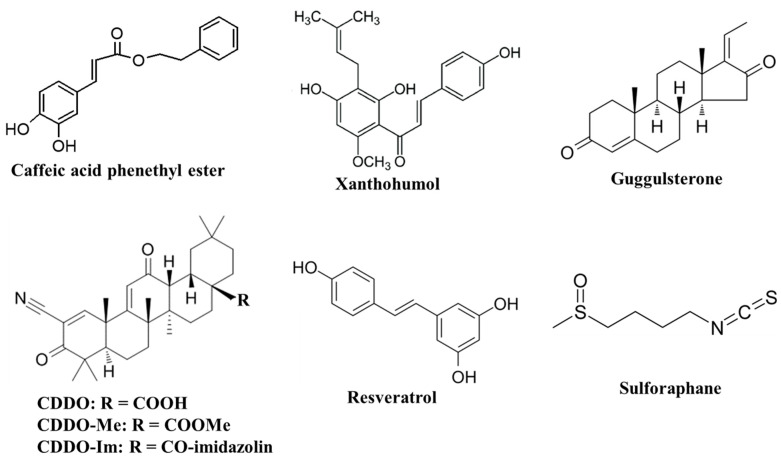
Structures of antioxidant phytochemicals with potential effects on autophagy and oxidative stress.

**Figure 2 molecules-27-04396-f002:**
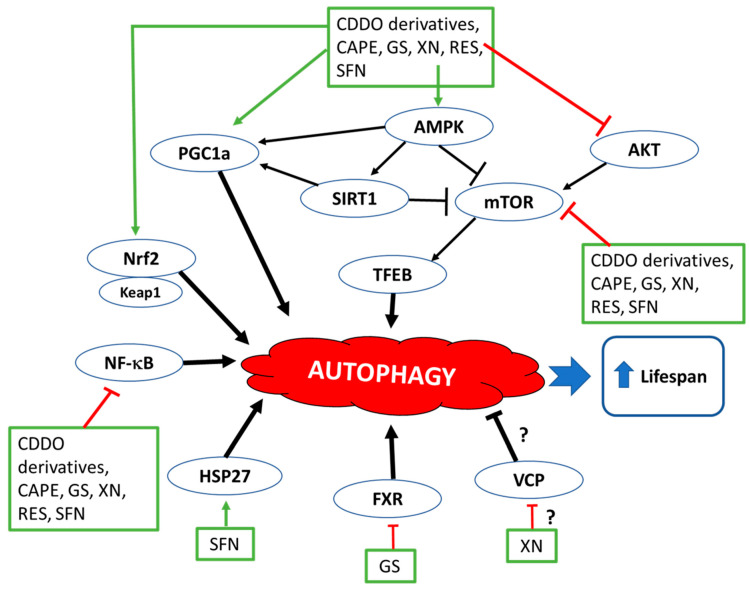
Regulation of autophagy by phytochemicals. Activation of AMPK signaling pathway promotes autophagy through the activation of multiple signaling molecules and transcription factors, including but not limited to SIRT1 and PGC1-a and the inhibition of mTOR [188]. mTOR further phosphorylates and inactivates TFEB, an important regulator of autophagy. Inhibitors of mTOR and its upstream pathway, Akt signaling, can promote autophagy [189]. Similarly, inhibitors of NF-κB have the potential to enhance the completion of autophagy process [190]. Nrf2, when bound to Keap1, a major regulator of Nrf2, is quickly degraded but under oxidative stress, it gets released from Keap1 to regulate autophagic process by getting translocated into the nucleus to activate the transcription of target genes [191]. Phytochemicals reviewed in this manuscript activate or inhibit multiple pathways to modulate autophagy.

**Figure 3 molecules-27-04396-f003:**
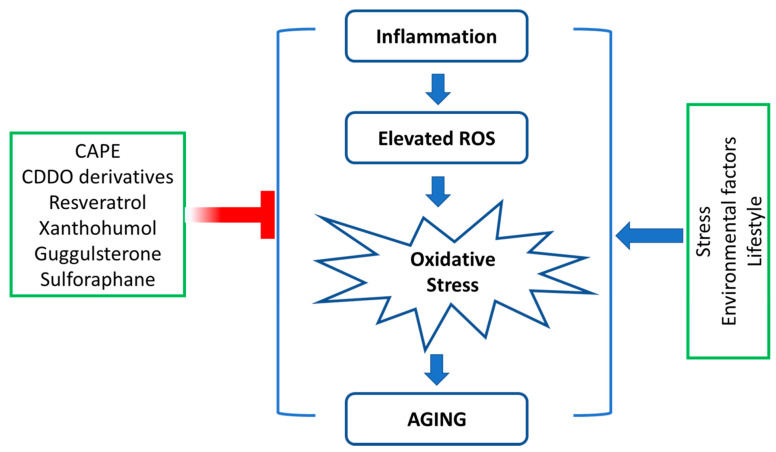
Phytochemicals inhibit OS-induced aging. Imbalance between the free radicals and antioxidants seen under inflammatory conditions leads to OS contributing to the process of aging. Several environmental factors and components of lifestyle play a significant role in increasing ROS levels, leading to oxidative stress-induced aging.

## Data Availability

Not applicable.

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
