# Peer review of "Antiaging Mechanism of Natural Compounds: Effects on Autophagy and Oxidative Stress"

_molecules, 2022, doi:10.3390/molecules27144396_

Round 1
Reviewer 1 Report
The authors presented the review on anti-aging mechanism of natural compounds and their derivatives. The content of this manuscript is well organized. This manuscript contains content that is of interest to experts in this field as well as non-experts. The manuscript has a merit to be published in Molecules. However, there are some suggestions which would improve the quality of the manuscript.
1. Why were the authors interested in this topic? Please describe the reason, along with its relevance to the authors' previous studies.
2. Has a review article similar to the content of this manuscript been published in the past? The authors should mention this point.
3. Is the content covered in this manuscript comprehensive for all papers reported on this topic so far? The authors should mention this point.
4. Are there any natural products with similar effects other than the compounds mentioned in this manuscript by the authors? And why did the authors only cover CDDO, CAPE, xanthohumol, guggulsterone, resveratrol, and sulforaphane in this manuscript? The authors should mention those points.
5. The authors should describe in detail the origins of CDDO, CAPE, xanthohumol, guggulsterone, resveratrol, and sulforaphane, the structural features that are believed to be involved in their activity, and other biological activities. The above information will be useful to the readers of this journal.
Author Response
Response to Reviewer #1 Comments
- Why were the authors interested in this topic? Please describe the reason, along with its relevance to the authors' previous studies.
Response 1: Thank you for your comment. We have interests in the beneficial effects of phytochemicals in a long term. Our previous studies have been focused on the cytoprotective and anti-cancer effects of selected natural compounds. Please refer to the following publications with PMID of 32304407, 30465316, and 30347819.
- Has a review article similar to the content of this manuscript been published in the past? The authors should mention this point.
Response 2: We have not published any review article similar to the content of this manuscript. We have provided up-to-date information on this topic.
- Is the content covered in this manuscript comprehensive for all papers reported on this topic so far? The authors should mention this point.
Response 3: Thank you for the comment. The content covered in this manuscript is comprehensive for papers reported on this topic since the year of 2000.
- Are there any natural products with similar effects other than the compounds mentioned in this manuscript by the authors? And why did the authors only cover CDDO, CAPE, xanthohumol, guggulsterone, resveratrol, and sulforaphane in this manuscript? The authors should mention those points.
Response 4: Thank you for your comments. We provided explanations in the revised manuscript why we picked those compounds from line 59 to 93 in red.
- The authors should describe in detail the origins of CDDO, CAPE, xanthohumol, guggulsterone, resveratrol, and sulforaphane, the structural features that are believed to be involved in their activity, and other biological activities. The above information will be useful to the readers of this journal.
Response 5: WE appreciate your comments. We have added detailed information regarding the origins and other biological activities of CDDO and its derivatives from line 348 to 359 in red. We have already included information regarding the origins and other biological activities of the other compounds. Therefore, no further information is provided.
Reviewer 2 Report
Despite the fact that the review article entitled " Anti-aging mechanism of natural compounds: effects on autophagy and oxidative stress" by Taylor et al. is an excellent review article and the topic has wide readership and interest, it cannot be published in its present form. My comments are divided into major and minor issues that need to be tackled before considering the article for publication.
MAJOR:
(1) It's not clear to me why the authors limited their review on these selected 6 compounds. Unless this is justified in the introduction, authors need to provide a comprehensive review covering key natural products with proven in vitro, in vivo anti-aging activities. As such, the current review overlooked/lacks the coverage of some important natural compounds with proven anti-aging activities such as:
- 7,8-dimethoxycoumarin (DMC), a natural coumarin that can be isolated from a variety of medicinal plants
Lee, N.; Chung, Y.C.; Kang, C.; Park, S.-M.; Hyun, C.-G. 7,8-dimethoxycoumarin Attenuates the Expression of IL-6, IL-8, and CCL2/MCP-1 in TNF-α-Treated HaCaT Cells by Potentially Targeting the NF-κB and MAPK Pathways. Cosmetics 2019, 6. [
- N-acetyl-5-methoxytryptamine (or phytomelatonin)
Ferri, F.; Olivieri, F.; Cannataro, R.; Caroleo, M.C.; Cione, E. Phytomelatonin regulates keratinocytes homeostasis counteracting aging process. Cosmetics 2019, 6.
- Several other compounds are also missing such as carnosic acid, curculigoside, curcumin, glycyrrhizic acid, mangiferin, mirkoin, asiaticoside, rosmarinic acid, tectorigenin, tyrosol, etc.
- Authors also need to provide evidence from skin studies both in in vitro and in vivo using keratinocytes and melanocytes as models to study anti-aging products and their effects on the key enzymes tyrosinase, hyaluronidase, elastase and collagenase.
(2) The review needs to include a section on the current and recent strategies and scientific advances for screening and investigating the anti-aging potential of natural products. For example, the natural products extracted from plant stem cells with anti-aging properties, which are similar to animal stem cells, and have been shown to be responsible for the growth and repair of damaged plant tissues.
Miastkowska, M.; Sikora, E. Anti-aging properties of plant stem cell extracts. Cosmetics 2018, 5.
MINOR:
-Abbreviate mechanism-of-action to (MOA) throughout the MS.
-L-195-196: The following statement is unclear and needs to be rephrased for clarity: "In this review, the interest of natural compounds that demonstrated organismal and cellular homeostasis were highlighted, which eventually facilitate autophagy and apoptosis."
-The chemical representation of ROS molecules formula need to be provided, such as H2O2 and HO., O2.- (The current online system does not permit me to show the free electrons for the later two, nor to provide subscript for O).
- Oxidative stress should be abbreviated to OS throughout the entire MS.
- As for the anti-aging mechanism-of-action for each of the listed plant-derived compounds, I suggest listing them in a table format detailing the MOA, molecular targets and cited references.
- L611: replace efficacious with effective.
-L421: correct anti-oxidants to antioxidants.
-L460: "Autophagy is a bulk, nonspecific protein degradation pathway that is involved in the 460
pathogenesis of cancer and neurodegenerative disease" . Meaning is not clear, you need to include words such as "to control the progression of" to show the positive and favorable impact of autophagy.
Author Response
Response to Reviewer #2 Comments
MAJOR:
(1) It's not clear to me why the authors limited their review on these selected 6 compounds. Unless this is justified in the introduction, authors need to provide a comprehensive review covering key natural products with proven in vitro, in vivo anti-aging activities. As such, the current review overlooked/lacks the coverage of some important natural compounds with proven anti-aging activities such as:
- 7,8-dimethoxycoumarin (DMC), a natural coumarin that can be isolated from a variety of medicinal plants
Lee, N.; Chung, Y.C.; Kang, C.; Park, S.-M.; Hyun, C.-G. 7,8-dimethoxycoumarin Attenuates the Expression of IL-6, IL-8, and CCL2/MCP-1 in TNF-α-Treated HaCaT Cells by Potentially Targeting the NF-κB and MAPK Pathways. Cosmetics 2019, 6.
- N-acetyl-5-methoxytryptamine (or phytomelatonin)
Ferri, F.; Olivieri, F.; Cannataro, R.; Caroleo, M.C.; Cione, E. Phytomelatonin regulates keratinocytes homeostasis counteracting aging process. Cosmetics 2019, 6.
- Several other compounds are also missing such as carnosic acid, curculigoside, curcumin, glycyrrhizic acid, mangiferin, mirkoin, asiaticoside, rosmarinic acid, tectorigenin, tyrosol, etc.
- Authors also need to provide evidence from skin studies both in in vitro and in vivo using keratinocytes and melanocytes as models to study anti-aging products and their effects on the key enzymes tyrosinase, hyaluronidase, elastase and collagenase.
Response to Major point (1): Thank you for your valuable comments. We provided explanations in the revised manuscript why we picked those compounds from line 59 to 93 in red. We added evidence from skin studies in the session of introduction (line 105 ~ 116), CDDO derivatives (line 417 ~ 433), CAPE (line 501 ~ 516), Xanthohumol (line 570 ~ 576), and Guggulsterone (line 614 ~ 622) in red.
(2) The review needs to include a section on the current and recent strategies and scientific advances for screening and investigating the anti-aging potential of natural products. For example, the natural products extracted from plant stem cells with anti-aging properties, which are similar to animal stem cells, and have been shown to be responsible for the growth and repair of damaged plant tissues.
Miastkowska, M.; Sikora, E. Anti-aging properties of plant stem cell extracts. Cosmetics 2018, 5.
Response to Major point (2): We appreciate the suggestions and would love to spend more time to discuss the current and recent strategies and scientific advances for screening and investigating the anti-aging potential of natural products in a different review paper. However, the focus of this manuscript is on the anti-aging effects of those selected natural compounds on autophagy and oxidative stress. We may not want to deviate from these main points.
MINOR:
-Abbreviate mechanism-of-action to (MOA) throughout the MS.
Response: We only mentioned mechanism of actions at one place throughout the whole paper. Therefore, no abbreviation is needed for this term.
-L-195-196: The following statement is unclear and needs to be rephrased for clarity: "In this review, the interest of natural compounds that demonstrated organismal and cellular homeostasis were highlighted, which eventually facilitate autophagy and apoptosis."
Response: We appreciate and agree with your comment. Therefore, we have replaced the original sentence with “In this review, we highlighted the anti-aging effects of those natural compounds that demonstrated organismal and cellular homeostasis through autophagy and apoptosis.” Please refer to line 251 and 252 in red.
-The chemical representation of ROS molecules formula need to be provided, such as H2O2 and HO., O2.- (The current online system does not permit me to show the free electrons for the later two, nor to provide subscript for O).
Response: Thank you for this comment. We added the information based on your suggestions from line 96 to 101 in red.
- Oxidative stress should be abbreviated to OS throughout the entire MS.
Response: Thank you for the comment. We have replaced osidative stress with OS through the entire manuscript.
- As for the anti-aging mechanism-of-action for each of the listed plant-derived compounds, I suggest listing them in a table format detailing the MOA, molecular targets and cited references.
Response: Thank you for the comment. Because we mainly focus on the anti-aging effects of those selected compounds on autophagy and oxidative stress, we did not describe other potential MOAs.
- L611: replace efficacious with effective.
Response: We appreciate the suggestion and have replace it accordingly in line 716.
-L421: correct anti-oxidants to antioxidants.
Response: We appreciate the suggestion and have correct it accordingly in line 494.
-L460: "Autophagy is a bulk, nonspecific protein degradation pathway that is involved in the pathogenesis of cancer and neurodegenerative disease" . Meaning is not clear, you need to include words such as "to control the progression of" to show the positive and favorable impact of autophagy.
Response: We agree with this comment and reword this sentence as “As a nonspecific protein degradation pathway, autophagy is designed to control the progression of cancer and neurodegenerative diseases” in line 550 and 551.
Round 2
Reviewer 2 Report
I'm satisfied with authors' response to my comments. My decision is to accept in current form. Good luck.